# The Effect of a 12-Week Physical Functional Training-Based Physical Education Intervention on Students’ Physical Fitness—A Quasi-Experimental Study

**DOI:** 10.3390/ijerph20053926

**Published:** 2023-02-22

**Authors:** Hailing Li, Jadeera Phaik Geok Cheong, Bahar Hussain

**Affiliations:** Faculty of Sports and Exercise Science, Universiti Malaya, Kuala Lumpur 50603, Malaysia

**Keywords:** fitness assessment, health growth, primary school, physical education and health curriculum model

## Abstract

Children have received much attention in recent years, as many studies have shown that their physical fitness level is on the decline. Physical education, as a compulsory curriculum, can play a monumental role in contributing to students’ participation in physical activities and the enhancement of their physical fitness. The aim of this study is to examine the effects of a 12-week physical functional training intervention program on students’ physical fitness. A total of 180 primary school students (7–12 years) were invited to participate in this study, 90 of whom participated in physical education classes that included 10 min of physical functional training, and the remaining 90 were in a control group that participated in traditional physical education classes. After 12 weeks, the 50-m sprint (F = 18.05, *p* < 0.001, $\eta_{p}^{2}$ = 0.09), timed rope skipping (F = 27.87, *p* < 0.001, $\eta_{p}^{2}$ = 0.14), agility *T*-test (F = 26.01, *p* < 0.001, $\eta_{p}^{2}$ = 0.13), and standing long jump (F = 16.43, *p* < 0.001, $\eta_{p}^{2}$ = 0.08) were all improved, but not the sit-and-reach (F = 0.70, *p =* 0.405). The results showed that physical education incorporating physical functional training can effectively promote some parameters of students’ physical fitness, while at the same time providing a new and alternative idea for improving students’ physical fitness in physical education.

## 1. Introduction

The essential foundation for a person to achieve good health is established during childhood, and this groundwork will subsequently determine health in adulthood [1,2]. A powerful marker of health in children is physical fitness (PF) [3], and this indicator appears to be growing in significance in their everyday lives [4]. Not only has PF been reported to be essential for performing school activities and meeting home responsibilities, but it has also been proclaimed to provide adequate energy for sports and alternative leisure activities [5]. There is evidence that children with low PF levels are associated with negatively impacting health outcomes, such as obesity, heart disease, impaired skeletal health, and poor quality of life [6].

Physical education (PE) is regarded as an ideal intervention point for promoting students’ health and PF because it involved almost all children [7]. Dobbins et al. [8] noted that PE-based interventions could ensure 100% of students were exposed to the intervention, which could benefit a large number of children across a wide range of demographic groups. Additionally, Errisuriz et al. [9] suggested that even minor PE modifications could improve fitness, and that the key was to discover a PE-based intervention that could be executed successfully. However, some studies have highlighted that there were barriers that prevented PE from playing a vital role in promoting students’ physical health and fitness, such as the scope, quantity, and quality of PE classes [10,11,12,13,14]. Ji and Li [15] also pointed out that PE in China had become a “safety class”, “discipline class”, and a “military class” which overemphasized the uniformity of movements and in which students often did not even sweat throughout the duration of the class. Such PE would not benefit public health and could make the students’ physique even worse [16]. In response to the shortcomings of traditional PE classes in China, a physical education and health curriculum model was proposed in 2015, which emphasized that each PE class must include 10 min of fitness training using diversified, enjoyable, and compensatory methods and means [17]. This model was mainly aimed at traditional PE classes that did not have a specific time allocated for PF exercises [16]. Many types of training methods have been suggested to improve PF such as school-based, high-intensity interval training [18], integrated neuromuscular exercise [19], game-based training [20], and sports training [21]. Moreover, functional training (FT) had also been advocated as a method to improve PF [22,23,24,25]. FT is a training concept and method system that focused on the basic posture and movement patterns, integrated various qualities to optimize the most basic movement abilities of the human body, and systematically optimized the links such as movement pattern, spinal strength, kinetic chain, recovery, and regeneration, to improve athletic ability [26].

FT is a relatively novel form of fitness [27], which originated in sports medicine, then was used in the coaching of sports, and was finally adopted in gymnasiums [28]. Nowadays, FT has become a fitness hot topic, ranking among the top 20 worldwide fitness trends based on the American Society of Sports Medicine (ACSM) global fitness trend survey since 2007 [29,30,31,32,33,34,35,36,37,38,39,40,41,42,43]. One of the reasons for its popularity is due to its health benefits; FT was designed to enhance the ability of exercisers to meet the demands of performing a wide range of activities of daily living at home, work, or play without undue risk of injury or fatigue [44]. Another reason was related to the performance benefit, as Boyle [45] noted that FT could help train speed, strength, and power for improved performance. Furthermore, FT required little space, little equipment, and little time, adding to its popularity [46]. In 2011, China introduced FT when preparing for the London Olympics [47]. To highlight the importance of FT in sports and distinguish it from medical institutions’ FT, the word “physical” was added before “functional training”, and physical functional training (PFT) became a widely used term to replace FT in China [48]. The PFT included pillar preparation, movement preparation, plyometrics, movement skills, strength and power, energy system development, and regeneration and recovery [26]. PFT had the characteristic of “separation and combination” in the application, so each PFT section could be designed and arranged flexibly, based on different stages of training and tasks, as needed [49].

With the deepening research on PFT in sports [25,50], more researchers began to transplant PFT to school PE. Through a systematic review of the research on PFT from 2009 to 2019, Kang, et al. [51] pointed out that researchers focused on PFT theoretical research from 2009–2012, applied research integrating PFT with PE from 2012–2014, and after 2014—with the enrichment and depth of PFT research topics—researchers focused on the application of PFT in school PE to improve students’ PF. However, these studies mainly involved teenagers and college students, with less attention on children [24,51]. Therefore, this research aimed to integrate PFT, an innovative PF training method, into PE and evaluate the impact of a 12-week PFT-based PE intervention on primary school students’ PF. The PFT intervention was designed to take up only 10 min of a regular PE lesson. It was hypothesized that the PF of the participants who underwent the PFT intervention would be improved after 12 weeks. Additionally, it was also hypothesized that the PF performance of the participants of the PFT group would be better than the participants of the control group at the end of the 12-week program.

## 2. Materials and Methods

### 2.1. Study Design

This study used a 12-week quasi-experimental design in which groups of participants were assigned to an intervention or control condition in a primary school in China. The intervention group participated in a 10-min PFT intervention program which was included in the PE class. The control group remained in the traditional PE class without the PFT intervention.

### 2.2. Participants

According to the PE and Health Curriculum Standards for Compulsory Education (2011 Edition) [52], the learning levels of primary school students were divided into three levels based on the characteristics of students’ psychosomatic development, which were first and second grades as level one, third and fourth grades as level two, and fifth and sixth grades as level three. Consequently, in this study, students from second grade, third grade, and sixth grade were selected to represent students from all three levels. Two classes from the selected grades were randomly chosen as the experimental class (EC) and control class (CC), respectively, with 30 students in each class. A total of 180 male and female students between the ages of 7 and 12 (8.97 ± 1.84 years) participated in the study.

All students read the participant information form, and their parents or guardian signed the informed consent form. This study was conducted according to the procedures approved by the University of Malaya Research Ethics Committee (UM.TNC2/UMREC—667, 19 November 2019).

### 2.3. Measurements

To evaluate the impact of PFT on students of different grades and levels, this study selected the mandatory PF indicators for all students based on the 2014 revised Chinese National Student Physical Fitness Standard (CNSPFS) battery [53] and were as follows: height and weight, 50-m sprint, sit-and-reach, and timed rope skipping. At the same time, two additional indicators of agility *T*-test and standing long jump were selected to evaluate agility and power, according to the PF test guidelines [54]. All measurements were taken before and after the 12-week intervention, in the same order.

#### 2.3.1. Height and Weight Test

Participants’ height and weight were measured by using a portable instrument (GMCS-IV; Jianmin, Beijing, China) to reflect their anthropometric characteristics. Testing was performed with the subject standing on the bottom plate of the equipment barefoot, with the head upright, the torso naturally straight, the upper limbs naturally drooping, and the heels close together. The toes were 60 degrees apart, and two to three seconds later, the measurement result appeared on the LCD [55]. The unit of measurement for height was in meters (m) and weight in kilograms (kg).

#### 2.3.2. 50-m Sprint Test

The 50-m straight racetrack, a starting flag, a whistle, and a stopwatch were used in this test, which was employed to assess speed. Before the test, the participants were in a ready position, standing with one foot in front of the other and the front foot behind the starting line. After the participants were prepared, the starter gave the instructions “set” then blew a whistle and waved the starting flag. The participants ran to the finish line as fast as possible while the finish line timer started timing, and the timekeeper stopped timing at the same time when the participant ran across the finish line. Each participant was allowed two trials. The best time was taken and recorded in seconds (s) to two decimal places.

#### 2.3.3. Sit-and-Reach Test

The sit-and-reach test was carried out by a seat-forward flexion tester (GMCS-IV; Jianmin, Beijing, China) to assess flexibility. During the test, the participant sat on a flat surface with legs straight and flat against the test longitudinal plate, approximately 10~15 cm apart. The upper body was bent forward, with the palms down and hands side by side, reaching forward along the measuring line as far as possible. Participants took the test twice, and the best result was recorded in centimeters (cm) to one decimal point.

#### 2.3.4. Timed Rope Skipping Test

The rope-skipping test was conducted by using a rope and a stopwatch to assess strength, muscle endurance, and coordination. During the test, participants were required to skip continuously for one minute with their feet together. The tester timed, counted, and recorded the number of times the rope was skipped.

#### 2.3.5. Agility *T*-test

A stopwatch, measuring tape, and four cones were used in this test to assess agility. Figure 1 shows the layout for the agility *T*-test. The participant began at cone 1, the same starting position for each trial. On the go command, the participant ran and touched cone 2, then cone 3. After touching cone 3, the participant shuffled sideways and touched cone 4. Next, the student shuffled back, touched cone 2, then ran back to the end line. Timing started on the command and stopped as the participant passed the end line. Each participant had two chances to take the best score in seconds (s).

#### 2.3.6. Standing Long Jump Test

The test was conducted by using a tape measure to assess power. During the test, with feet slightly apart, the participant stood behind a line drawn on the ground. A two-foot takeoff and landing were used, with forwarding force provided by swinging the arms and bending the knees. The participant attempted to jump as far as possible, landing on both feet without falling back. The test outcome was measured from the start line to the closest point of contact (back of the heel) after landing. Two jumps were allowed, and the best was taken in cm.

### 2.4. Intervention Program

The program included three stages, starting with two weeks of the basic stage, which was mainly used to learn the basic movement pattern, then moving on to five weeks of advanced stage Ⅰ, and another five weeks of advanced stage Ⅱ.

The basic stage focused on teaching the basic movement patterns to develop PF based on mastering basic movement patterns. Advanced stage Ⅰ comprised of PFT modules using the medicine ball, agility ladder, pad, or cone to develop the participants’ PF. Advanced stage Ⅱ was mainly based on the same PFT modules of stage Ⅰ but with an increase in the training load. In terms of arranging the training load, it was generally to overcome the self-weight and light load. The change of load from advanced stage Ⅰ to advanced stage Ⅱ was realized through the following forms: (1) the change of training route, from unidirectional to multidirectional change, and (2) the distance and repeat times. The exercise components and a detailed arrangement of the intervention are presented in Table 1.

### 2.5. Procedures

First, in this study, a team of research assistants comprised of three primary school teachers from the experimental school was trained in data collection and intervention implementation.

Then, the teachers organized the participants to perform the height and weight test, followed by the 50-m sprint, sit-and-reach, timed rope skipping, agility *T*-test, and standing long jump test for the baseline assessment. During the tests, PE teachers first put forward some safety considerations to the participants. After the introduction, they used 10 min to organize the students to warm up, including jogging and muscle stretching before taking the baseline tests. In the testing process, each student had two opportunities for each test, and the best score was recorded.

Next, participants were required to attend three PE sessions per week for 12 weeks. The EC took part in the PE class that was incorporated with 10 min PFT program while the CC participated in the traditional PE classes that had no mandatory requirements for PF training [16] and were mostly comprised of games activities (see Table 1 for example of games activities).

Finally, all participants were tested again by using the same format as the baseline.

### 2.6. Statistical Analysis

SPSS 25.0 software (IBM SPSS Statistics for Windows, Version 25.0. IBM Corp.: Armonk, NY, USA) was used to process and analyze the PF test results of children. The normality distribution of data was checked by using the Shapiro–Wilk test for all measurements. Based on the distribution results, the independent sample *T*-test (parametric) or the Mann–Whitney U test (nonparametric) was used to compare the test scores between the EC and CC prior to the start of the experiment. The paired sample *T*-test (parametric) or Wilcoxon signed-rank test (nonparametric) was used to compare the score changes between baseline and posttest, for the EC and CC, respectively. Cohen’s d was used to describe effect sizes for the parametric test according to the following conventions: small (0.20 to 0.49), medium (0.50 to 0.79), and large (0.80 and above) (Cohen, 1988). Pearson’s r was used to describe effect sizes for the nonparametric tests according to the following conventions: small (0.10 to 0.29), medium (0.30 to 0.49), and large (0.50 and over) [56,57].

Analysis of covariance (ANCOVA) was conducted to determine significant differences between the posttest scores of EC and CC. Height, weight, and baseline scores of each measurement variable were entered as covariates. Quade’s rank-transformed analysis of covariance (nonparametric ANCOVA) as an alternative method was used if the data did not meet the assumption for ANCOVA [58,59]. Effect sizes for statistically significant outcomes were reported as partial eta squared ($\eta_{P}^{2})$, with small, medium, and large effect sizes classed as 0.01, 0.06, and 0.14, respectively [56].

## 3. Results

### 3.1. Comparison of Baseline Characteristics

An overview of the anthropometric characteristics of participants is shown in Table 2. There were no significant differences between the EC and CC at baseline for all measures in all grades (*p* > 0.05) (Table 3).

### 3.2. Effect of Intervention

After 12 weeks of PE classes, within-group comparisons were made between participants in both the experimental and control classes at each grade level (see Table 4). For the second grade, the EC showed significant improvement in the 50-m sprint, timed jump rope, agility *T*-test, and standing long jump after the experiment (*p* < 0.001), whereas scores for the sit-and-reach (*p* = 0.187) were not significant. The CC showed significant improvements in the 50-m sprint, timed rope skipping, and standing long jump after the experiment (*p* < 0.001), whereas the scores for the sit-and-reach (*p =* 0.073) and agility *T*-test (*p =* 0.670) were not significant. In third grade, there was a significant increase in all indicators in both the EC and CC (*p* < 0.001). In the sixth grade, there was a significant increase in the posttest values compared to the baseline of all indicators in the EC (*p* < 0.001), whereas in the CC, there was a nonsignificant increase in the timed rope skipping (*p =* 0.483) and standing long jump (*p =* 0.171) and a significant increase in the other indicators (*p* < 0.05). Although the results varied by grade level, overall, participants in both EC and CC made significant improvements in PF scores after 12 weeks of PE classes (*p* < 0.05).

The results of the comparison between the experimental and control class groups are shown in Table 5. Overall, the differences in the postintervention indicators between the students in EC and CC were highly significant, except for the sit-and-reach (*p =* 0.405). The specific results for each grade were as follows. In the second grade, EC was significantly better than CC in the 50-m sprint, timed rope skipping, and agility *T*-test, but the differences in sit-and-reach (*p =* 0.680) and standing long jump (*p =* 0.079) were not statistically significant. In the third grade, the 50-m sprint, timed rope skipping, and agility *T*-test scores of EC were significantly better than CC, whereas the differences in sit-and-reach (*p =* 0.120) and standing long jump (*p =* 0.244) between the two groups were not statistically significant. In the sixth grade the 50-m sprint, timed rope skipping, and standing long jump scores of EC were significantly better than CC, whereas the differences in sit-and-reach (*p =* 0.980) and agility *T*-test (*p =* 0.222) indicators between the two groups were not statistically significant.

## 4. Discussion

The purpose of this study was to evaluate the impact of a 12-week PFT-based PE intervention on primary school students’ PF. It was hypothesized that the PF of the participants who underwent the PFT intervention would be improved after 12 weeks. In addition, it was hypothesized that the PF of the participants of the PFT group would be better than the participants of the control group at the end of the 12-week intervention.

When the baseline scores were compared with the post-test scores, the results revealed that the PF of the EC students who participated in the PFT intervention had improved after 12 weeks, in line with our hypothesis. At the same time, students of the CC had also significantly improved across time. It appeared that the traditional PE class, which comprised mostly of games activities, was able to improve students’ PF after 12 weeks, regardless of whether there was a 10-min PFT component included in the class or not. This is a positive finding for PE in schools—the current classes were somewhat beneficial to the students. This finding was supported by Cocca, et al. [60], Cocca, et al. [61], and Petrušič, et al. [62] who also found that PE classes, including games, could improve the PF of students.

When baseline data were entered as covariates, the results of this study showed that there was a significant difference in the scores of the EC over the CC in all PF variables except for the sit-and-reach test, which also supported our hypothesis that participants of the EC would display better PF performance compared to the CC. The results of the study suggested that PFT could provide a novel exercise method for PE modules to improve students’ PF.

In this study, the largest differences between the groups were in the 50-m sprint, which evaluated speed, and the timed rope skipping, which assessed muscle strength and coordination. The EC at each level was significantly better than the CC. This was consistent with previous studies showing that PFT could improve muscle strength and speed. Yildiz, Pinar, and Gelen [24] implemented an eight-week FT versus traditional training program in preteen tennis players (9.6 ± 0.7 years) and reported that FT was more effective than traditional training in both strength and speed. Tomljanović, Spasić, Gabrilo, Uljević, and Foretić [27] similarly proved that a five-week functional training program for males aged 23 to 25 could improve speed and strength performance. Limited literature is available to compare the combined effect of PFT on coordination. Nevertheless, Li et al. [63] pointed out that PFT emphasized the integration of nerve-muscle functions and strengthens the efficient control of nerves over muscles in multiple dimensions, all-around range, and speed in a wide range, which helped speed, agility, and coordination gain better performance.

Next, the agility *T*-test showed significant differences between the EC and CC at levels one and two. In the third level, although both the EC and CC improved over time, there was no significant difference between the two classes. The positive changes in measured agility might be related to enhanced lower-extremity reflexes and proprioception and improved postural control in subjects through 12 weeks of training [27]. Meanwhile, the insignificant results of the students in level three might be related to the students’ 50-m × 8 shuttle run practice, which also promoted the development of the CC students’ agility in the corresponding teaching and practice.

In addition, the standing long jump test, which evaluated power, showed no significant differences between the groups in levels one and two but revealed a significant difference in level three. This may be related to the motor coordination ability that affected explosive power [27]. Low-level students are not as good as high-level students in postural control and muscle coordination during movement practice. The stimulation generated during movement practice might not be enough to stimulate the neuromuscular system to burst intensity [64]. Therefore, students in the lower levels could not benefit from PFT until they reached a later age, at which point motor coordination was better developed.

Finally, there was no significant difference between EC and CC in all three levels of the sit-and-reach test for assessing flexibility. According to previous research [23,24,65], PFT interventions could significantly improve the flexibility of participants. It was possible that the inconsistency of the results with other studies could be because the PFT program of this study did not include dynamic stretching and static stretching exercises, which were often arranged in warm-up and cool-down modules of training programs [26,66]. Because this study mainly focused on the main model of the PE class, the stretching module was not included in the PFT program.

In summary, the highlight of this study was that primary school students’ PF, such as speed, coordination, strength, and agility, was superior after 10 min of PFT in each PE class, which was in line with the previous studies that had found that PFT could improve PF [22,45,67,68]. It was possible that PFT emphasized the neural involvement in the training process [63,69,70] to affect the entire neuromuscular system [69,71]. In addition, according to a previous study, PFT also strengthened the body’s stretch reflex, which increased the reflexivity of muscle activity through the rapid pulling of the muscle shuttle to promote muscle force and power output [26]. However, it was also found that students at the lower levels were less effective than those at the higher levels in terms of power generation, which could be related to the quality of the movement performed. PFT focused on the quality of the movement rather than the load and quantity of the movement [26]. Hence, in lower-level students who had weaker limb control, the quality of their movement could have been affected, and consequently their performance was worse than the upper-level students.

There are also limitations in this study. First, the participants were all primary school students, which had a limited cognitive level, the quality of the movement completion was affected to a certain extent in the process from understanding the movement to implementing it. Secondly, in the selection of movements in the program, some simple and easy-to-implement movements were selected, which reduced the intensity of the exercise to a certain extent. Finally, all the students came from one class at each level. This was to accommodate the timetable, as all of the PE lessons were not conducted for all of the students at the same time. As such, we chose the participants from one class in each level based on the available slot given by the PE teacher.

## 5. Conclusions

The study results identified that the 12-week PE effectively improved the PF level of students both in EC and CC. However, the PFT integrated into PE produced more positive effects on students’ PF than traditional PE, such as speed, agility, and coordination, which revealed that PFT could be an acceptable and effective type of exercise for school children to improve their PF. The targeted selection of PFT exercises which were designed to be incorporated into the existing PE modules could be adopted by PE teachers to develop students’ PF.

## Figures and Tables

**Figure 1 ijerph-20-03926-f001:**
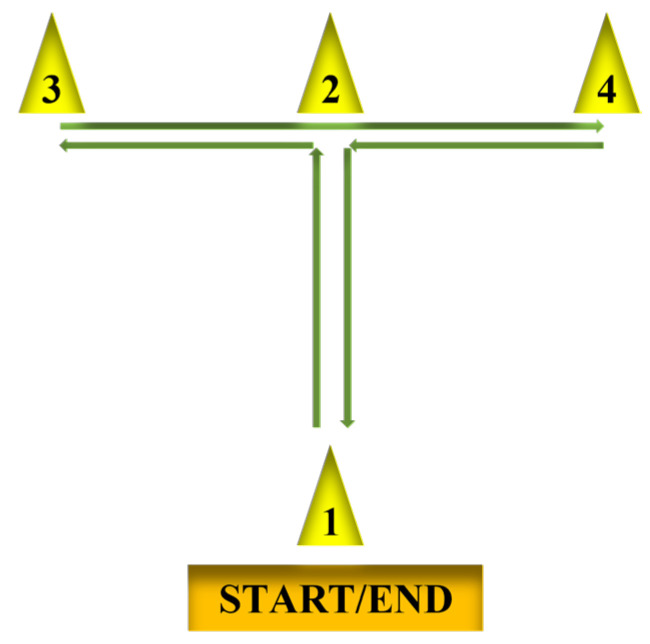
Agility *T*-test procedure.

**Table 1 ijerph-20-03926-t001:** Details of a 10-min physical fitness intervention for the experimental class and examples of corresponding activities of the control class.

Week	Session	Experimental Class					Control Class	
		Module	Exercise	Set	Rep ^1^	Rest (s)	* Games	Set
1	1	Movement pattern	Squats; lateral squats (switch); jump squats; squat turn 90°(L&R) ^2^.	2	6–8	30	Gather quickly ^7^	2–3
	2		Squats; lateral squats (switch); jump squats; squat turn 90°(L&R).	3	6–8	30	Find a partner ^8^	2–3
	3		Jump squats; bounds; squat turn 180° (L&R).	3	6–8	30	Gather quickly	2–3
2	4	Movement pattern	Forward lunge (L&R); lateral lunge (L&R); backward lunge (L&R).	2	6–8	30	Find the partner	2–3
	5		Walking lunges; walking lunges+ rotation; backward walking lunges; backward walking lunges + rotation.	2	6–8	30	Birds fly ^7^	2–3
	6		Seated step; stride; carioca; trot; bound; sprint (6 m).	3	-	30	Eagle catching chicks ^8^	2–3
3	7	Plyometric training	Hop back and forth; lateral hop; one leg hop back and forth (L&R); one leg lateral hop (L&R); jack jump; cross jump (8 s–10 s).	2	-	15	Yangtze River, Yellow River ^8^	2–3
	8		Forward jumps; one leg forward jump; skipping; “Z” ^3^ bounds (10 m).	2	-	30 s	Eagle catching chicks	2–3
	9		Forward jumps + sprint; one leg forward jump + sprint; skipping + sprint; “Z” bounds + sprint (10 m)	2	-	30 s	Barter race ^9^	2–3
4	10	Strength and power (medicine ball)	Squats with a ball; lateral squats with a ball; lateral squats with a ball; squats with a ball pushing forward; squats with a ball pushing up; lateral squats with a ball pushing forward; lateral squats with a ball pushing up.	2	6–8	30	Dashing through the trenches ^10^	2–3
	11		Forward, lateral, and backward squats with a ball (L&R); forward, lateral, and backward squats with a ball pushing forward (L&R); forward, lateral, and backward squats with a ball pushing up (L&R).	2	6–8	30	Relay race ^9^	2–3
	12		Diagonal chop down; diagonal chop up; deadlift with a ball; squat jumping with a ball; bound with a ball; squat jumping turn 90° with ball.	2	6–8	30	Long throw race ^10^	2–3
5	13	Strength and power (pad)	Plank; supination—abdominal crunch; plank—hand touch shoulder; supination-reverse crunch (10 s–20 s).	3	-	30	Dashing through the trenches	2–3
	14		“V” ^4^ sits with rotation; supination-leg rotation; pronation-hyperextension; supination—45-degree abdominal crunch; supination-leg raise; hyperextension (10 s–20 s).	2	-	30	Frog crossing the river ^11^	2–3
	15		Glute bridge; supination—arm overhand sit with straight legs; supination-leg raise; glute bridge—one leg lift (L&R); V-up (10 s–20 s).	2	-	30	Barter race	2–3
6	16	Movement skill (agility ladder)	One foot in each rung; two feet in each rung; high knees; forward carioca; in–in–out–out (8 m).	3	-	30	Sandbag throwing ^10^	2–3
	17		One foot in each rung; Two feet in each rung; high knees; forward carioca; in–in–out–out (lateral) (8 m).	3	-	30	Hunters hitting ducks ^10^	2–3
	18		One foot in each rung + sprint; two feet in each rung + sprint; high knees + sprint; forward carioca + sprint; in–in–out–out + sprint (3 × 8 m).	2	-	30	Target shooting game ^10^	2–3
7	19	Energy systems development (cone)	5–10 shuttle verbal commands (30 m).	2–3	-	60	Hit dragon tail ^10^	2–3
	20		5–10–15 shuttles (60 m).	2–3	-	60	Beautify the campus ^9^	2–3
	21		5–10–15–20 shuttles (100 m).	2–3	-	60	Rolling ball relay ^9^	2–3
8	22	Plyometric training (agility ladder)	Hops; “Z” hops; hopscotch two feet in; hopscotch one foot in (8 m).	3	-	30	Big Fish Net ^8^	2–3
	23		One-leg hop (L&R); one leg “z” hop (L&R) (8 m).	2	-	30	Beautify the campus	2–3
	24		Lateral hop; forward and backward lateral hop; one leg lateral hop (L&R); one leg forward and backward hop (L&R) (8 m).	2	-	30	Frog crossing the river	2–3
9	25	Strength and power (medicine ball)	Squats with a ball; lateral squats with a ball; squats with a ball pushing forward; squats with a ball pushing up; lateral squats with a ball pushing forward; lateral squats with a ball pushing up.	3	6–8	30	Target shooting game	2–3
	26		Walking lunges pushing forward; walking lunges pushing up; walking lunges with rotation (10 m).	2	-	30	Long throwing game	2–3
	27		Walking lunges pushing forward; walking lunges pushing up; walking lunges with rotation (10 m).	3	-	30	Hunters hitting ducks	2–3
10	28	Strength and power (pad)	Plank; supination—abdominal crunch; plank—hand touch shoulder; supination-reverse crunch (10 s–20 s).	3	-	30	Hit dragon tail	2–3
	29		“V” ^4^ sits with rotation; supination-leg rotation; pronation-hyperextension; supination—45-degree abdominal crunch; supination-leg raise; hyperextension (10 s–20 s).	3	-	30	Hunters playing ducks	2–3
	30		Glute bridge; supination—arm overhand sit with straight legs; supination—leg raise; glute bridge—one leg lift (L&R); V-up. (10 s–20 s)	3	-	30	Target shooting game	2–3
11	31	Movement skill (cone)	Sprint; side slide; shuttles; “S” running; running circle; lateral running circle (10 m).	3	-	30	Sandbag throwing	2–3
	32		“L” ^5^ sprint; “L” sprint + side slide; “L” sprint + carioca running; “V” sprint; “V” sprint + side slide; “V” + “8” circle running (10 m).	2	-	30	Target shooting game	2–3
	33		Square sprint; square side slide + sprint; square carioca + sprint + side slide; square running circle; “X” ^6^ sprint; agility box (verbal commands). (10 m)	2–3	-	30	Hunters playing ducks	2–3
12	34	Energy systems development (cone)	5–10–15 shuttles (60 m).	2–3	-	60	Dashing through the trenches	2–3
	35		5–10–15–20 shuttles (100 m).	2–3	-	60	Beautify the campus	2–3
	36		5–10–15–20–25 shuttles (150 m).	2–3	-	60	Hit dragon tail	2–3

* Games, the activities used by the school teachers, denote examples of activities that were carried out during the traditional PE class; ^1^ repetition, ^2^ left & right,^3^ Z pattern, ^4^ V pattern, ^5^ L pattern, ^6^ X pattern, ^7^ student formation dismissal-resume game, ^8^ chasing games, ^9^ racing games, ^10^ throwing games, ^11^ jumping games.

**Table 2 ijerph-20-03926-t002:** Descriptive statistics for participants’ anthropometric characteristics.

Grade	Gender	Experimental Class				Control Class			
		Height (cm)	Weight (kg)	Age (years)	N	Height (cm)	Weight (kg)	Age (years)	N
		(Mean ± SD)	(Mean ± SD)	(Mean ± SD)		(Mean ± SD)	(Mean ± SD)	(Mean ± SD)	
2nd	Male	127.59 ± 6.79	26.68 ± 5.56	7.31 ± 0.48	16	130.53 ± 6.04	30.39 ± 5.95	7.40 ± 0.51	15
	Female	126.26 ± 4.85	25.75 ± 3.93	7.00 ± 0.00	14	128.18 ± 7.27	26.61 ± 4.66	7.00 ± 0.00	15
3rd	Male	136.11 ± 5.10	32.56 ± 6.54	8.39 ± 0.50	18	139.26 ± 5.92	33.03 ± 7.14	8.26 ± 0.45	19
	Female	134.25 ± 3.60	26.42 ± 4.52	8.25 ± 0.45	12	134.68 ± 5.62	27.95 ± 4.26	8.36 ± 0.51	11
6th	Male	144.89 ± 4.00	36.25 ± 4.53	11.18 ± 0.39	17	149.26 ± 9.12	42.81 ± 9.74	11.53 ± 0.52	15
	Female	149.09 ± 4.98	37.08 ± 4.61	11.31 ± 0.48	13	147.95 ± 6.87	40.02 ± 9.74	11.60 ± 0.51	15
ALL	Male	136.37 ± 8.78	31.94 ± 6.76	8.98 ± 1.69	51	139.65 ± 10.15	35.21 ± 9.18	9.00 ± 1.80	49
	Female	136.33 ± 10.68	29.73 ± 6.76	8.82 ± 1.89	39	137.16 ± 10.89	31.88 ± 9.21	9.05 ± 2.07	41

**Table 3 ijerph-20-03926-t003:** Students’ PF characteristics during the baseline test.

Grade	Variable	Experimental Class		Control Class		*p*
		Mean	SD	Mean	SD	
2nd	50-m sprint	12.26	0.96	12.16	1	0.690
	Timed rope skipping	87.7	31.25	81.43	26.88	0.408
	Sit-and-reach	11.78	4.24	10.45	5.79	0.315
	Agility *T*-test	13.75	1.44	13.24	1.27	0.146
	Standing long jump	124.47	15.73	127.30	14.98	0.478
3rd	50-m sprint ^1^	10.5	0.62	10.42	0.62	0.362
	Timed rope skipping	142.97	11.99	143.57	16.7	0.874
	Sit-and-reach	11.01	5.37	10.91	7.51	0.951
	Agility *T*-test	10.25	0.69	10.35	0.59	0.562
	Standing long jump	131.73	11.5	132.73	13.75	0.761
6th	50-m sprint	9.07	0.66	8.95	0.64	0.480
	Timed rope skipping	136.23	26.04	131.07	27.78	0.460
	Sit-and-reach	10.49	5.75	12.95	6.55	0.128
	Agility *T*-test	10.91	0.8	11.23	0.71	0.112
	Standing long jump ^1^	162.83	18.18	158.53	21.28	0.193
All	50-m sprint	10.61	1.51	10.51	1.52	0.657
	Timed rope skipping ^1^	122.30	34.63	118.69	36.14	0.364
	Sit-and-reach	11.09	5.13	11.44	6.67	0.699
	Agility *T*-test ^1^	11.64	1.84	11.60	1.51	0.633
	Standing long jump ^1^	139.68	22.62	139.52	21.68	0.980

^1^ Mann–Whitney U test

**Table 4 ijerph-20-03926-t004:** Within-group changes by grade after the intervention.

Grade	Group	Variable	Mean	SD	*p*	Effect Size		95% CI
						*d ^2^*	*r ^3^*	
2nd	EC ^4^	50-m sprint	9.89 ↓	0.60	0.000 * ^1^	−2.96	-	(−3.693, −2.228)
		Timed rope skipping	122 ↑	16.92	0.000 *	1.37	-	(0.803, 1.927)
		Sit-and-reach	12.32 ↑	3.66	0.405	-	−0.11	(−0.105, −0.115)
		Agility *T*-test	11.03 ↓	0.57	0.000 *	−2.48	-	(−3.157, −1.810)
		Standing long jump	134.27 ↑	13.32	0.000 *	0.67	-	(0.152, 1.193)
	CC ^5^	50-m sprint	10.83 ↓	1.29	0.000 *	−1.15	-	(−1.699, −0.606)
		Timed rope skipping	99.07 ↑	26.66	0.000 *	0.66	-	(0.139, 1.179)
		Sit-and-reach	11.48 ↑	5.20	0.073	0.19	-	(−0.320, 0.694)
		Agility *T*-test	13.15 ↓	0.97	0.670	−0.08	-	(−0.586, 0.427)
		Standing long jump	132.07 ↑	16.06	0.000 *	0.31	-	(−0.202, 0.816)
3rd	EC	50-m sprint	10.00 ↓	0.51	0.000 *	−0.88	-	(−1.411, −0.351)
		Timed rope skipping	160.37 ↑	10.26	0.000 *	1.56	-	(0.981, 2.137)
		Sit-and-reach	12.95 ↑	5.44	0.000 *	-	−0.60	(−0.570, −0.625)
		Agility *T*-test	9.79 ↓	0.34	0.000 *	−0.85	-	(−1.374, −0.318)
		Standing long jump	140.63 ↑	10.38	0.000 *		−0.62	(−0.589, −0.646)
	CC	50-m sprint	10.29 ↓	0.61	0.000 *	−0.21	-	(−0.719, 0.296)
		Timed rope skipping	150.67 ↑	17.62	0.000 *	-	−0.53	(−0.506, −0.554)
		Sit-and-reach	12.33 ↑	7.79	0.000 *	0.19	-	(−0.322, 0.693)
		Agility *T*-test	10.11 ↓	0.52	0.000 *	−0.43	-	(−0.944, 0.080)
		Standing long jump	138.8 ↑	13.89	0.000 *	-	−0.62	(−0.590, −0.646)
6th	EC	50-m sprint	8.52 ↓	0.63	0.000 *	−0.85	-	(−1.381, −0.324)
		Timed rope skipping	161.27 ↑	24.47	0.000 *	0.99	-	(0.455, 1.527)
		Sit-and-reach	14.01 ↑	6.64	0.000 *	0.57	-	(0.051, 1.083)
		Agility *T*-test	10.58 ↓	0.71	0.000 *	−0.44	-	(−0.948, 0.076)
		Standing long jump	172.53 ↑	18.25	0.000 *	0.53	-	(0.018, 1.047)
	CC	50-m sprint	8.75 ↓	0.68	0.013 *	−0.3	-	(−0.812, 0.206)
		Timed rope skipping	136.23 ↑	26.04	0.483	0.19	-	(−0.316, 0.699)
		Sit-and-reach	16.39 ↑	6.81	0.000 *	0.51	-	(0.000, 1.029)
		Agility *T*-test	10.69 ↓	0.89	0.000 *	−0.67	-	(−1.191, −0.151)
		Standing long jump	153.93 ↓	18.72	0.171	−0.23	-	(−0.737, 0.278)
All	EC	50-m sprint	9.47 ↓	0.89	0.000 *	-	−0.60	(−0.587, −0.605)
		Timed rope skipping	147.88 ↑	25.72	0.000 *	-	−0.58	(−0.571, −0.589)
		Sit-and-reach	13.09 ↑	5.37	0.000 *	-	−0.43	(−0.423, −0.436)
		Agility *T*-test	10.47 ↓	0.76	0.000 *	-	−0.55	(−0.545, −0.562)
		Standing long jump	149.14 ↑	22.02	0.000 *	0.42	-	(0.129, 0.720)
	CC	50-m sprint	9.96 ↓	1.27	0.000 *	-	−0.45	(−0.441, −0.455)
		Timed rope skipping	128.66 ↑	32.12	0.000 *	-	−0.33	(−0.330, −0.340)
		Sit-and-reach	13.40 ↑	6.95	0.000 *	-	−0.47	(−0.460, −0.475)
		Agility *T*-test	11.32 ↓	1.55	0.000 *	-	−0.29	(−0.287, −0.296)
		Standing long jump	141.60 ↑	18.59	0.001 *	-	−0.25	(−0.242, −0.250)

^1^ * indicates that the value is significant at *p* ≤ 0.05, ^2^ effect size for the paired-samples *t*-test, ^3^ effect size for the Wilcoxon signed-rank test, ^4^ experimental class, ^5^ control class, ↑ indicates that the value increases, and ↓ indicates that the value decreases.

**Table 5 ijerph-20-03926-t005:** Changes between-group by grade after the intervention.

Grade	Variable	∆Mean ^3^	SE	95% CI	df	F	*p*	$\eta_{P}^{2}$
2nd	50-m sprint ^2^	−13.25	3.39	(−20.03, −6.47)	1	15.29	0.000 * ^1^	0.21
	Timed rope skipping ^2^	13.4	3.04	(7.31, 19.48)	1	19.41	0.000 *	0.25
	Sit-and-reach	−0.26	0.62	(−1.49, 0.98)	1	0.17	0.680	0.00
	Agility *T*-test	−2.28	0.2	(−2.67, −1.88)	1	132.28	0.000 *	0.71
	Standing long jump ^2^	5.12	2.79	(−0.46, 10.70)	1	3.37	0.072	0.06
3rd	50-m sprint ^2^	−7.98	2.02	(−12.02, −3.93)	1	15.57	0.000 *	0.21
	Timed rope skipping ^2^	10.83	2.59	(5.64, 16.02)	1	17.46	0.000 *	0.23
	Sit-and-reach	0.7	0.45	(−0.19, 1.60)	1	2.50	0.120	0.04
	Agility *T*-test ^2^	−7.4	2.02	(−11.43, −3.37)	1	13.49	0.001 *	0.19
	Standing long jump ^2^	3.05	2.17	(−1.29, 7.38)	1	1.98	0.165	0.03
6th	50-m sprint ^2^	−7.84	3	(−13.85, −1.83)	1	6.81	0.012 *	0.11
	Timed rope skipping	17.41	2.87	(11.75, 23.06)	1	36.84	0.000 *	0.17
	Sit-and-reach ^2^	0.61	2.16	(−3.71, 4.94)	1	0.08	0.778	0.00
	Agility *T*-test ^2^	3.15	3.39	(−3.62, 9.93)	1	0.87	0.356	0.02
	Standing long jump ^2^	13.19	2.92	(7.34, 19.03)	1	20.41	0.000 *	0.26
All	50-m sprint ^2^	−20.43	4.81	(−29.92, −10.94)	1	18.05	0.000 *	0.09
	Timed rope skipping ^2^	26.11	4.95	(16.35, 35.87)	1	27.87	0.000 *	0.14
	Sit-and-reach ^2^	3.16	3.79	(−4.31, 10.64)	1	0.70	0.405	0.00
	Agility *T*-test ^2^	−21.07	4.13	(−29.22, −12.92)	1	26.01	0.000 *	0.13
	Standing long jump ^2^	15.74	3.88	(8.08, 23.41)	1	16.43	0.000 *	0.08

^1^ * indicates that the value is significant at *p* ≤ 0.05, ^2^ Quade’s rank-transformed ANCOVA test, ^3^—the difference between experimental class and control class.

## Data Availability

The data presented in this study are available on request from the corresponding author.

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
