# Peer review of "The Effect of a 12-Week Physical Functional Training-Based Physical Education Intervention on Students’ Physical Fitness—A Quasi-Experimental Study"

_ijerph, 2023, doi:10.3390/ijerph20053926_

Round 1

Reviewer 1 Report

This is an interesting study that incorporated 10 minutes of physical functional training (PFT) into primary school PE classes and compared changes in fitness with regular PE classes. Overall, I think the study would be of interest to other researchers studying physical functional training and PE among youth as well as PE practitioners. The article is mostly well-written, although it does need some grammatical editing for clarity. I will first list my key concerns (all of which I believe are addressable), followed by more detailed comments by section.

Key concerns:

First, the stated study hypothesis (Lines 82-84) is not what you actually tested in your study. In it, you state differences between groups as a “higher level of PF,” and yet you tested differences in the changes of PF over the intervention period. So, your main analysis results (Lines 246-259) do not match your stated hypothesis as they are on changes in versus levels of PF.

Second, since you randomized intact classes, you need to account for the clustering of participants within your models since the entire class got the intervention. This can happen using multi-level modeling, but it is highly likely that your study is underpowered as a result.

Third, Figure 2 does not add to your paper beyond what’s already presented in your tables. And, the text in all of your figures is too small to read without zooming in quite a bit to the page. Either use a table or figure, but not both – and my preference is for the tables. If you retain Figure 3, you need to explain what the third asterisk is for.

Comments by section:

Introduction

Line 43 – be consistent with abbreviating PE.

Methods

Study design – was this study conducted in one or more than one school? And was the study conducted in Malaysia or China?

Measurements (Line 112) – the reference citation given here [52] for the PF testing guidelines does not directly link to the information stated in your paper.

Line 141 – What do you mean by “after getting off their feet” here?

 Table 1 – I appreciate the details you provided in the table, but I do not think others could replicate what you did without discussing it further with you. Later, you also state that the games listed in the table were part of the regular PE class, which is also indicated by the CC column. So, it would be helpful to change the table title to clarify this as the first time through I thought the table was only representing the experimental condition. Finally, please explain all acronyms (EC, CC) at the bottom of the table.

2.5 Procedures – what training, if any, did the PE teachers receive for the intervention and the assessments? Did research staff members conduct any of the classes or measurements or was it just the teachers? And how many total teachers were part of the study?

Results

Table 2. Please add in participant sex. Explain EC and CC below the table.

Table 3. Explain EC and CC below the table.

Table 4. If this table represents changes from pre- to posttest, that should be clearly indicated with a + or – symbol before each mean. From what’s presented, it is impossible to tell the direction of the change. Also, sprint is misspelled as sprit throughout the table.

Table 5. The symbol following Time rope skipping and SLJ (and more throughout the table) is not explained below the table.

Discussion

Lines 262-269 – The explanation of the study purpose here does not match what you had earlier. Please reconcile all to be the same.

Line 281 – You again mention your “hypothesis” and what you have here is different than what your actual hypothesis was.

Line 283 – What do you mean by “interesting”?

Line 285 – What do you mean by “most outstanding results”?

Lines 296, 304 – “getting improved” is not grammatically correct to use here.

Lines 305-306 – Grammar is incorrect in this sentence.

Lines 321-342 – Be careful to not overreach in your summary statements by stating things that you did not actually measure (e.g., the movement conditioned reflexes could be consolidated. Etc.).

Conclusion

It is important to note that the PE only group also improved in physical fitness.

Author Response

Thank you very much for your valuable comments and suggestions. Our responses can be found in the attached file.

Reviewer 2 Report

Introduction - is pretty well presented. I have two suggestions for the authors:

1. to present more studies similar or close to the topic regarding the PF and PE of students (there are few studies listed, especially in children - because the age range analyzed in this study is 7-12 years)

2. highlight very clearly, what was the novelty of this study.

Keywords: it is recommended the authors not to repeat these words in the title of the article, i.e. physical education, physical functional training; physical fitness.

Methods

Lines 90-93 - ''Both groups underwent baseline testing and a post-test was conducted at the end of the 12-week intervention''...ok, but you must present what were these tests in order to understand very clearly all the readers.

„A total of 180 male and female students between the ages of 7 and 12.......” but what was the initial number of subjects? was it a bigger number and there were 180 left in the end? were there refusals or children who did not pass the initial tests? please fill with the necessary information.

Research Ethics Committee (UM.TNC2/UMREC – 667) – please insert which was the approval date of the study.

From Games:

Gather

quickly

Find a

partner

Birds fly

Eagle

catching

chicks etc .....is there any description in the article or on additional information of these means used?

For SPSS 25.0 software, please enter the country and city, for example: "IBM Corporation, Armonk, NY, USA".

Results - they are quite well presented.

Additional, please provide the full name of the abbreviations used in the tables, i.e. SLJ, EC, CC ETC.

Discussions - the authors should also present what were the limits of this study.

Author Contributions: the font used by the authors is different, please use Palatino Linotype

References - use journal abbreviations, not their full names.

Author Response

We are grateful for all you comments and suggestions. Our responses are in the attached file. Thank you

Round 2

Reviewer 1 Report

The paper has been improved by the authors based on responses to the prior critique.

I appreciate the updated hypothesis, but as written it neither justifies the design nor the statistical analysis used in the paper. In essence, you tested both within and between group differences for 180 students clustered within three grade levels with two conditions per level. Yet, you only state that the PFT interventions would have a “positive impact” on students’ PF. This must be reconciled to better justify your design and statistical analysis used.

I understand how the analyses were conducted as explained, however, you still did not account for the clustering of students within the classes in your analysis. I guess the editors can decide whether they see that as an issue or not. In any regard, it should be stated as a limitation of your study.

In the discussion, lines 329-338, you still are over-reaching with your wording in comparison to what you actually measured. It is appropriate to state that it is possible those were the underlying mechanisms for the improvements in PF but you cannot definitively state that was the case.

Finally, multiple word choice/grammatical errors still exist throughout the paper that should be carefully copy edited to fix.

Author Response

Q1. I appreciate the updated hypothesis, but as written it neither justifies the design nor the statistical analysis used in the paper. In essence, you tested both within and between group differences for 180 students clustered within three grade levels with two conditions per level. Yet, you only state that the PFT interventions would have a “positive impact” on students’ PF. This must be reconciled to better justify your design and statistical analysis used.

A1. Thank you very much for your suggestion, we have further refined our hypotheses as follows: It was hypothesized that the PF of the participants who underwent the PFT intervention would be improved after 12 weeks. Additionally, it was also hypothesized that the PF performance of the participants of the PFT group would be better than the participants of the control group at the end of the 12-week program.

Q2. I understand how the analyses were conducted as explained, however, you still did not account for the clustering of students within the classes in your analysis. I guess the editors can decide whether they see that as an issue or not. In any regard, it should be stated as a limitation of your study.

A2. Thank you very much for your valuable opinion. When we carried out the research, we could not conduct the intervention among students of the same grade from different classes at the same time due to the school's time table. Therefore, in order to carry out the research smoothly, we randomly select a class corresponding to each level to conduct research. We did not account for the clustering of students as we were uninformed about the issues that you have raised. We apologise for our oversight and based on your suggestion, we have stated this as a  research limitations.

Q3. In the discussion, lines 329-338, you still are over-reaching with your wording in comparison to what you actually measured. It is appropriate to state that it is possible those were the underlying mechanisms for the improvements in PF but you cannot definitively state that was the case.

A3. Thank you for your comments. We have changed the sentences to suggest that the reasons we provided in our discussion were a possibility.

Q4. Finally, multiple word choice/grammatical errors still exist throughout the paper that should be carefully copy edited to fix.

A4. Thank you very much for your opinions, we used Grammarly to make further corrections to the article, and proof read the article again.
